# *"I could hang up if the practitioner was a prat"*: Australian men's feedback on telemental healthcare during COVID-19

Zac E. Seidler[1,2,3]*, Michael J. Wilson[1,2], John L. Oliffe[4,5], David Kealy[6], John S. Ogrodniczuk[6], Andreas Walther[7], Simon M. Rice[1,2]

1 Orygen, Parkville, Melbourne, Australia, 2 Centre for Youth Mental Health, The University of Melbourne, Melbourne, Australia, 3 Movember, Melbourne, Australia, 4 School of Nursing, University of British Columbia, Vancouver, Canada, 5 Department of Nursing, The University of Melbourne, Melbourne, Australia, 6 Department of Psychiatry, University of British Columbia, Vancouver, Canada, 7 Clinical Psychology and Psychotherapy, University of Zurich, Zurich, Switzerland

* zac.seidler@orygen.org.au

**Data Availability Statement:** Data cannot be shared publicly because of ethical restrictions. Data can be provided upon written request to the corresponding author, or to HumanEthics-

## Abstract

The COVID-19 pandemic restrictions, uncertainties and management inconsistencies have been implicated in men's rising distress levels, which in turn have somewhat normed the uptake of telemental healthcare services (i.e., phone and/or video-conference-based therapy). Given past evidence of poor engagement with telemental health among men, this mixed-methods study examined Australian men's use of, and experiences with telemental health services relative to face-to-face care during the pandemic. A community sample of Australian-based men ($N = 387$; age $M = 47.5$ years, $SD = 15.0$ years) were recruited via Facebook advertising, and completed an online survey comprising quantitative items and open-response qualitative questions with the aim of better understanding men's experiences with telemental healthcare services. In total, 62.3% ($n = 241$) of participants reported experience with telemental health, and regression analyses revealed those who engaged with telemental health were on average younger, more likely to be gay and university educated. Men who had used telemental health were, on average, more satisfied with their therapy experience than those who had face-to-face therapy. Among those who had telemental healthcare, marginally lower satisfaction was observed among regional/rural based relative to urban men, and those who had to wait longer than 2 months to commence therapy. Qualitative findings highlighted positive aspects of telemental healthcare including comfort with accessing therapy from familiar home environments and the convenience and accessibility of telemental health alongside competing commitments and COVID-19 restrictions. Conversely, drawbacks included technical limitations such as crosstalk impeding therapeutic progress, disconnects and audio-visual lag-times and the 'impersonal' nature of telemental healthcare services. Findings broadly signal COVID-19 induced shifts norming of the use of virtual therapy services, with clear scope for improvement in the delivery of therapeutic practice using digital modalities, especially among help-seeking men.

Enquiries@unimelb.edu.au. Provision of data will be subject to ethics approval for the intended secondary use by researchers who meet criteria for access to confidential data.

**Funding:** ZS received funding for this survey from Movember: https://au.movember.com/ The funders had no role in study design, data collection and analysis, decision to publish, or preparation of the manuscript.

**Competing interests:** The authors have declared that no competing interests exist.

## Introduction

As the COVID-19 pandemic spread across the globe in early 2020, significant social, economic and political pressures accompanied the spread of the virus [1]. Foremost among these were lockdowns and social distancing measures which had unprecedented and profound implications for men's mental health linked to increased levels of distress [2–4]. In response to the COVID-19 restrictions and the widespread mental health challenges, telemental healthcare services increased, as the sector attempted to meet surging demand [5, 6].

Telemental healthcare is characterized by the provision of services remotely via telephone or videoconferencing, including intake, assessment and psychotherapy [7]. Within an Australian context, telemental health has long been purported as the future of innovative mental healthcare as a means to adapt to the country's vast and diversely populated geographies [8]. Nonetheless, resistance to telemental health was driven by a lack of experience or training with this virtual mode of service delivery, the clinical traditions of being there physically in consultations and the uncertainty surrounding clinical effectiveness, as well as financial disincentive to provide virtual services due to lack of government funding support [9, 10]. While there was minimal telemental health service utilisation in rural and remote areas leading up to 2020 (a notable exception being the special provision of mental health services to individuals affected by Australian bushfires in 2019–20) [11], it was the outbreak of COVID-19 that saw the treatment modality become mainstream across Australia. Incentivized by adjusted financial rebates from the Australian Government, telemental healthcare services rapidly grew [12]. The uptake of this healthcare shift was evident by June 2020, with one in five Australian adults engaging in a telemental health service in the past month, notwithstanding concerns about sub-group inequities for accessing such technologies [11, 13]. Indeed, the 'new normal' of COVID-19 has precipitated a rapidly shifting milieu espousing the broad appeal of telemental health for inclusively caring for a broad range of individuals.

Provision of men's mental healthcare services has grappled with meeting the needs of a population with traditionally poorer uptake [14], engagement [15], and retention in treatment compared to women [16, 17]. Improving men's mental health treatment is particularly pressing in the COVID-19 context given that men make up 75% of suicide deaths in Australia, and that prominent risk factors for male suicidality––including unemployment, relationship breakdowns, financial hardship and social isolation––have been amplified during the pandemic [18–20]. Among commonly cited barriers to mental healthcare access amongst men are restrictive opening hours, fixed service locations, waiting rooms, time-sensitive consultations, need for transportation and taking time off work [16, 21]. Many of these barriers may be ameliorated by telemental healthcare services [22]. Herein autonomy, agency and privacy afforded by online environments and telemental healthcare can effectively lever those masculine priorities to norm men's engagement with services [23, 24].

There are of course caveats and cautions to espousing telemental health as the utopia and tonic for men. Across general medicine, findings from an Australian-based cohort during COVID-19 highlighted men experienced telemental health services as significantly worse than face-to-face care, however these findings lacked exploration of specifically which aspects of telemental health services failed to engage men [6]. Others report sex differences, such as a study of over one million Californians indicating that compared to men, women were significantly more likely to choose to engage with telemental health over face-to-face care [25]. Grounding sex differences in telemedicine is limited in determining the gendered dimensions of men's engagement with telemental health. Indeed, to the authors' knowledge, there is no published research reporting men's experiences with tele*mental* health. This is a significant knowledge gap given the unprecedented and ongoing mental health challenges accompanying

the COVID-19 pandemic [1]. The current study reports Australian-based men's experiences of telemental healthcare services in comparison to face-to-face therapy. Using a mixed-method survey methodology, this study aimed to investigate demographic predictors of access to and satisfaction with telemental health relative to face-to-face services, and then to complement this data with a qualitative exploration of the helpful factors and areas needing improvement related to delivery of the telemental health modality for men.

## Method

### Design

The present study employed a cross-sectional design, involving a mix of quantitative survey items and open-response qualitative questions. This mixed-methods survey design, used in similar, recent surveys in the male help-seeking population, provides a viable avenue to explore both the predictors of and/or relationships between discrete phenomena alongside inductive assessment of their quality, in a large group of participants. The inclusion of open-response questions in primarily quantitative surveys has grown as an adjunct to gain depth and breadth of insight [26]. Amidst a mental health system that struggles to engage male clients [15, 27, 28], this methodology allows a sequential use of mixed methods to be inclusive of men's narratives about their experiences to extend what is reported through quantitative survey items.

### Participants and procedure

Following ethics approval from the University of XX Human Ethics Sub-Committee (ethics ID: XX), participants were recruited between 25th October-29th December 2021 via targeted social media advertisements mirroring the approach applied in our previous survey of men's experiences of help-seeking (see [17]). Australian men aged ≥16 years (i.e., inclusive of mature minors) were invited to take part in a brief online survey about their mental health during the COVID-19 pandemic. Data presented here are a subset of the larger survey, hosted by Qualtrics. The scope of the ethical approval granted for this study limits public availability of study data. Study data will be provided following written request to the corresponding author, or to HumanEthics-Enquiries@unimelb.edu.au. Provision of data is subject to ethical approval for the subsequent use of study data.

Participants who clicked through the advertisement link were immediately presented with a plain language statement and consent form, where informed consent was obtained via a yes/no survey item. Consenting participants were then asked to work through the ~15-minute survey, which contained a range of quantitative and free text entry items exploring mental health and help-seeking experiences throughout the COVID-19 pandemic. Participants were given the option to enter the draw for a $500 voucher as compensation for their time.

### Measures

**Help-seeking.** Following the collection of demographic data (see Table 1), a series of quantitative items derived from similar previous surveys (e.g. [17]) assessed whether participants had sought help from a mental health professional, either in their lifetime or during the COVID-19 pandemic (e.g. since March 2020; see S1 Appendix for full list of items). If help-seeking during the pandemic was endorsed, further items assessed the length of wait time prior to an initial appointment, alongside the frequency and format of sessions (e.g. face-to-face or telemental health). In addition, participants who had received therapy during COVID-19 were asked whether they were still working with their therapist.

**Table 1. Univariate comparisons between users of telemental health and only face-to-face therapy.**

| | Telemental health | Face-to-face only | χ² / t | p, ES |
|---|---|---|---|---|
| | **n = 241** | **n = 146** | | |
| Age: M (SD) | 45.5 (14.7) | 51.0 (14.9) | **3.56** | **< .001, .37** |
| Sexuality: % (n) | | | | |
| Heterosexual | 62.2 (150) | 76.7 (112) | **8.92** | **.030, .15** |
| Gay | 27.0 (65) | 15.8 (23) | | |
| Bisexual | 8.3 (20) | 5.5 (8) | | |
| Other | 2.5 (6) | 2.1 (3) | | |
| Education level: % (n) | | | | |
| University educated | 66.0 (159) | 48.6 (71) | **11.35** | **.001, .17** |
| Non-university educated | 34.0 (82) | 51.4 (75) | | |
| Place of residence: % (n) | | | | |
| Urban | 73.4 (177) | 65.8 (96) | 3.59 | .108, .08 |
| Regional/rural | 26.6 (64) | 34.2 (50) | | |
| Employment status: % (n) | | | | |
| Employed | 70.1 (169) | 65.1 (95) | 1.07 | .301, .05 |
| Unemployed | 29.9 (72) | 34.9 (51) | | |
| First time help-seeker: % (n) | | | | |
| Yes | 24.5 (59) | 23.3 (34) | 0.07 | .790, .01 |
| No | 75.5 (182) | 76.7 (112) | | |
| Wait time to commence therapy: % (n) | | | | |
| A few days—2 weeks | 48.5 (117) | 45.9 (67) | 2.07 | .355, .07 |
| 2 weeks—2 months | 37.8 (91) | 34.9 (51) | | |
| >2 months | 13.7 (33) | 19.2 (28) | | |
| Still in therapy: % (n)* | | | | |
| Yes | 65.9 (145) | 45.2 (61) | **14.75** | **< .001, .20** |
| No | 34.1 (75) | 54.8 (74) | | |
| Satisfaction with therapy: M (SD) | 47.4 (13.0) | 42.6 (14.5) | **3.17** | **.002, 0.34** |

Note. *cell counts do not sum to equal column totals due to missing data for this variable.

**Satisfaction.** The Satisfaction with Therapy and Therapist Scale-Revised (STTS-R) [29] was used to measure participants' overall satisfaction with their therapy experience. This scale includes 12 items (e.g., "I am satisfied with the quality of the therapy I received") with Likert-type response scales ranging from 1 (strongly disagree) to 5 (strongly agree). Responses are summed, with higher scores indicating greater satisfaction with therapy. The internal consistency reliability of the scale is excellent (α = .93) [29] with comparable reliability observed here (α = .97).

**Experiences with telemental health.** In this survey, the term 'telemental health' referred to both virtual (video) and phone-based (audio only) sessions. Participants who endorsed that some or all of their sessions throughout COVID-19 were conducted via telemental health were also asked what proportion of their sessions with their most recent practitioner were conducted via telemental health, in addition to whether participants preferred telemental health or face-to-face therapy, and whether participants would opt to see a practitioner via telemental health in future, given the option.

Finally, for those participants who reported having engaged with telemental health since the outset of the COVID-19 pandemic, two open-ended, free-text questions were used to further explore the elements of their telemental health experience that they liked and disliked

most: 1) *What did you like about seeing a mental health practitioner via telemental health (what was good about it)?* 2) *What did you not like about seeing a mental health practitioner via telemental health (what was bad about it)?*

## Data analysis

**Quantitative analysis.** All analysis was completed in SPSS 27. Firstly, univariate proportions of variables pertaining to participants' experiences with telemental health were examined for the full sample. Next, a binary variable was created for participants who had any sessions via telemental health throughout the pandemic (coded 1), and those that only had face-to-face therapy (coded 2). Sociodemographic variables were compared using this variable to investigate any differences in the characteristics of those participants who engaged with telemental health vs face-to-face therapy only, alongside items pertaining to participants' experiences of help-seeking (i.e., whether it was their first time; the wait time they experienced; and whether they were still in therapy at the time of survey completion; and their satisfaction with care). Finally, the sample was filtered such that only those participants who had any telemental health since March 2020 were selected, and differences in their satisfaction with care were examined across demographic characteristics and items pertaining to help-seeking experiences (i.e., whether it was their first time; the wait time they experienced; and whether they were still in therapy at the time of survey completion; and their openness to receiving telemental health in future) via multiple linear regression.

All categorical analyses involved chi-square tests, where effect sizes were examined according to Cramer's V ($\varphi c$), where 0.1 is considered a small effect, 0.3 a medium effect and 0.5 a large effect. For *t*-tests, effect sizes were examined using Cohen's d where 0.2 reflects a small effect, 0.5 a medium effect and 0.8 a large effect.

**Qualitative analysis.** Responses to the qualitative questions assessing participants' experiences of telemental health were analysed using inductive thematic analysis, across a six-stage process of coding and theme development [30]. First, familiarisation with the data involved an in-depth read and re-read of the full dataset. Responses were subsequently copied to a spreadsheet for initial independent coding by the first author (ZS). Codes were initially developed to encompass similar responses and labelled descriptively, following a process of organisation and subsuming under higher-order categories in consult with the second author (MW). For example, the codes '*feeling less judged*' and '*greater sense of control*' were subsumed into the subtheme '*Greater psychological/physical safety*'. Once consensus was reached that the thematic structure accurately represented the dataset, all authors reviewed the overall data and findings to further revise the results reported here. Also, in the writing up of the qualitative results, ZS and MW adjusted the thematic labels and illustrative quotes.

## Results

Among the sample, 62.3% (*n* = 241) reported receiving some or all therapy sessions via telemental health during the pandemic. Among those who reported experience with telemental health, it was most common for participants to have received all of their sessions via telemental health (48.4%, *n* = 105), where the remainder reported a mix between telemental health and face-to-face therapy. For those men who received any treatment via telemental health, 24.5% (*n* = 59) were seeking help for the first time during the pandemic. Regarding preferences for telemental health or face-to-face therapy, a majority of participants preferred face-to-face (65.1%, *n* = 149), whereas only 12.7% (*n* = 29) stated a preference for telemental health. Nevertheless, when asked if they would continue to work with a practitioner via telemental health in future, more noted that they would (43.2%, *n* = 99), than would not (33.6%, *n* = 77). Overall,

60.2% ($n$ = 145) of those who had telemental health reported they were (at time of survey completion) still seeing the same mental health practitioner, relative to 34.1% ($n$ = 75) who had discontinued therapy.

## Comparing participants who had any telemental health versus only face-to-face therapy

As depicted in Table 1 below, those who engaged with therapy via telemental health during COVID-19 were, on average, younger. In addition, gay men were more likely to have engaged with telemental health than heterosexual men, alongside university-educated participants relative to non-university educated participants. There were no other differences between those who received telemental health, and those who did not, on any other socio-demographic markers including whether it was their first experience of seeking help, wait time to commence therapy, current place of residence, or employment status. Participants who experienced telemental health were more likely to still be in therapy at the time of survey completion relative to those who only had face-to-face therapy. Finally, those who engaged with telemental health were, on average, more satisfied with their therapy relative to those who only had face-to-face therapy, with a small effect size observed.

## Exploring satisfaction with therapy among those who had telemental health

Finally, among participants who had experience with telemental health, a multiple linear regression analysis was conducted to examine any subgroup differences in levels of satisfaction with therapy via telemental health. The overall model was significant, ($F$(10,209) = 2.859, $p$ = .002), where $R^2$ = .12. Regional or rural participants had significantly lower satisfaction with telemental health relative to urban participants. Aside from this, no demographic variables emerged as significant factors differentiating levels of satisfaction with telemental health. Regarding variables pertaining to participants' help-seeking, satisfaction with telemental health was unrelated to whether participants were seeking help for the first time. However, a significant difference in satisfaction according to wait time was observed, where those who waited more than 2 months to commence therapy experienced significantly less satisfaction with therapy via telemental health on average, relative to those who waited a few days– 2 weeks (see Table 2).

## Qualitative findings

Two hundred and fifteen participants who had used telemental health services since March 2020 provided open response inputs to separate questions exploring participants' likes and dislikes about seeing a mental health practitioner via telemental health. Themes derived from the data were broadly structured around helpful aspects of telehealth and areas needing improvement. Though the themes are presented as discrete, all participants were asked to respond to both questions. The thematic breakdown for each question is provided below and a relative count is offered in Table 3.

**Telemental health: The benefits.**   Two broad themes were developed from the data regarding the benefits of telemental healthcare services. These ranged from specific elements of telemental health which engendered feelings of comfort and safety for participants, to the comparative convenience and accessibility in telemental health compared with traditional face-to-face healthcare interactions.

*Comfort with therapy*. Fifty-five participants suggested that they engaged in telemental health treatment from the comfort of their own home or an environment of their choice,

**Table 2. Multiple linear regression of satisfaction with therapy according to demographic and help-seeking predictors, among those who had telemental healthcare.**

| Predictor (reference category) | *B* | 95% CI for *B* Lower | 95% CI for *B* Upper | *SE* | *t* | *p* | Partial correlation |
|---|---|---|---|---|---|---|---|
| Age | 0.10 | -.02 | .23 | 0.06 | 1.68 | .094 | .12 |
| Place of residence (urban) | | | | | | | |
| Regional/rural | -4.06 | -7.89 | -0.24 | 1.94 | -2.09 | **.037** | -.14 |
| Sexuality (heterosexual) | | | | | | | |
| Gay | 0.06 | -3.85 | 3.96 | 1.98 | 0.03 | .977 | .01 |
| Bisexual | -2.56 | -9.31 | 4.20 | 3.43 | -0.75 | .457 | -.05 |
| Other sexuality | -6.32 | -16.68 | 4.03 | 5.25 | -1.20 | .230 | -.08 |
| Employment status (employed) | | | | | | | |
| Unemployed | -0.59 | -4.43 | 3.26 | 1.95 | -0.30 | .764 | -.02 |
| Education level (non-university educated) | | | | | | | |
| University educated | 1.75 | -1.88 | 5.38 | 1.84 | 0.95 | .344 | .07 |
| First time help-seeking (yes) | | | | | | | |
| No | 3.34 | -0.60 | 7.28 | 2.00 | 1.67 | .096 | .12 |
| Wait time (a few days—2 weeks) | | | | | | | |
| 2 weeks—2 months | 0.35 | -3.32 | 4.01 | 1.86 | 0.19 | .852 | .01 |
| >2 months | -8.36 | -13.57 | -3.15 | 2.64 | -3.16 | **.002** | -.21 |

which resulted in a far less confronting experience relative to face-to-face healthcare interactions. The physical comfort of being in familiar surroundings (e.g., bedroom; car) rather than what was described as a cold and often stressful 'medical setting' had flow-on effects. One participant noted there were "*fewer external distractions, can focus better*" (606) while others noted the virtual setting imbued a relaxed and open nature to their subsequent treatment:

**Table 3. Thematic representation and count of qualitative findings across positive and negative aspects of telemental health.**

| Theme | Subtheme | Count |
|---|---|---|
| **Telemental health: The benefits** | | |
| Comfort with therapy | Less confronting and more comfortable | 55 |
| | Greater psychological safety | 18 |
| | Just enjoyed basic therapy | 16 |
| | Confidentiality & anonymity | 9 |
| Convenience and accessibility | No need to compromise for therapy | 73 |
| | Telemental health is timely, easy and accessible | 47 |
| Nothing/not sure | | 30 |
| **Telemental health: The drawbacks** | | |
| Therapy interrupted | Technical difficulties or limitations | 32 |
| | Blurred boundaries between home, work and therapy | 31 |
| | Greater difficulty accessing and sharing emotions | 20 |
| Disaffection and distance from therapist | Impersonal, lacked connection and rapport | 127 |
| | Missing non-verbal cues, body language and feedback | 36 |
| Nothing/not sure | | 33 |

Note. *n* = 26 responses were missing from each question (i.e., benefits and drawbacks)

*Felt like I had to prepare less. Could be at home in pyjamas, comfortable in my room, on the phone. Rather than waking up earlier and getting ready for an appointment. Sometimes felt a bit less 'superficially composed' too, meaning not having to dress up and sit in a medical room helped me open up a bit more. (110*)

This comfort was often linked to feelings of control given the "*inherent distancing that comes with telehealth*" (448), with another participant noting "*I could hang up if the practitioner was a prat*!*" (335).*

Eighteen participants described how this sense of comfort and control underpinned feelings of psychological and/or physical safety in the telemental health setting. The lower risk of contracting COVID-19 was relayed by some participants with "ensured protection" (369) and "health safety" (626), improving their comfort in accessing therapy in a pandemic context. Additionally, others reported how when they felt most vulnerable and in need of therapy, having the choice to seek treatment from home allayed their fears and even improved engagement:

*I feel safe at home which sometimes makes it easier to open up—can still attend when not feeling the best, less missed sessions (193).*

Nine participants also noted they liked feeling as though their telemental health treatment was confidential, private and anonymous:

*I like the idea that the person doesn't not know me and we are unlikely to ever cross paths (177)*

This was uniquely important for one participant living in a smaller regional community:

*Maintained some professional distance when living and working in small regional community (274)*

*Convenience and accessibility*. Seventy-three participants suggested that the best part of their telemental health experience was the convenience of being able to schedule and engage in treatment at a time and place of their choosing, reducing the burden of travel time and potential for missing, most commonly, work commitments.

*The unique convenience of being able to fit in mental health appointments amongst other personal and work commitments. Not having to travel to see mental health practitioners (148)*

Forty-seven participants reflected on the "*ease of access*" (85) and timely nature of telemental health with shortened wait times for an appointment meaning "*my options have opened up immensely*" (48) as treatment was more readily available. This was especially true for participants in regional and rural settings with restricted access to specialist mental health professionals:

*The wait times are shorter via telehealth. The wait times to see someone in my home town are extremely long. There is also no one in my home town who specialises in uncommon issues, e. g. autism and paraphilias. But there are lots of people who specialise in that via telehealth. (25)*

Many participants also noted the benefits of telemental health in the COVID-19 context, where barriers such as the need for screening or testing pre-appointment were removed. This meant that participants were "*able to do the session even if I was isolated*" (594) or still attend therapy "*during hard lockdown*" (509). Importantly, accessibility was often lamented as "*about the only advantage*" (40) to telemental health and thus responses acknowledging accessibility were not always positive in nature. Some participants noted "*there was simply no alternative at the time.*" (678)

**Telemental health: The drawbacks.** Two broad themes were developed to reflect areas requiring improvement in the extent to which telemental health services effectively engage men. These were grouped around the notion of disconnection from therapy, where aspects of telemental health stymied the extent to which participants could engage with therapy. Participants also described various forms of disconnection from therapists, with the telemental health modality conferring difficulties regarding building rapport, often alongside missing non-verbal cues.

*Therapy interrupted.* Thirty-two participants noted some frustrations with the technical limitations of telemental health, with interrupted internet service impacting a sense of flow and fluidity in session. This disjointedness led to "*lots of interjecting over each other*" (9) for one participant and "*caused distress and anxiety*" (369) for another. One participant conveyed frustration regarding the requisite level of digital competence required to engage with therapy via telemental health:

> *Having to walk providers through technical issues is extremely frustrating. Then going onto mobile phone instead. . .Generally it sucked (241)*

Specific elements of therapy (or specific therapies) were deemed untenable in an online environment, including eye movement desensitization and reprocessing. Participants also reported missing certain aspects of face-to-face therapy that served to facilitate engagement, where one participant missed the "*use of white board for explanations and workarounds*" (353). Finally, four participants worried about data security and privacy with the fear "*that companies are transcribing and recording my patient data, responses and emotions. . .*" (27).

While accessing therapy from home presented advantages in terms of comfort for some participants, thirty-one participants spoke of the difficulty in navigating blurred boundaries between their private world (e.g., home and work) and the therapy space. The lack of privacy and barriers to openness in the environment, such as "*having to speak quietly as there were family members*" (24) and "*therapist seeing my living space*" (179) were disconcerting. One participant described how therapy became "*slightly reminiscent of every other zoom meeting*" (575), likely reflecting the COVID-19 context and widespread directives to work from home. Moreover, while some liked the convenience of not having to travel, participants here noted the importance of "*separation in my brain*" (85) with processing time to organise thoughts and learnings:

> "*There is a beneficial ritualism travelling to and from a face-to-face session. It allows for introspection both before and after.*" (83).

Twenty participants added that an overarching discomfort with the telemental health modality limited their ability to authentically open up, as "*it was awkward and difficult to accurately convey and read emotions*" (7). One participant reflected that not being in the room together led him to judge his own responses, further highlighting a sense of unease with exploring one's internal world without the physical presence of a therapist:

*Could always think of things afterwards that I wished I had said. I felt like saying some things on the phone sounded pathetic or needy, but don't sound that way in a face-to-face situation. (638)*

*Disaffection and distance from the therapist.* One hundred and twenty-seven participants described telemental health as lacking the therapeutic qualities they had come to expect, with one participant lamenting "*it's dehumanising*" (309). These participants disliked what they experienced as the "*impersonal*" (109) nature of telemental health, with difficulty building rapport as it "*didn't feel real*" (169), there was "*less connection*" (404), was "*very distant and detached from my therapist*" (152), and they in turn, "*felt less supported*" (656). Responses here conveyed the profound effects that a lack of shared physical space with one's therapist can have on therapeutic engagement.

Additionally, thirty-six participants reported the inability to read non-verbal cues via telemental health as limiting their engagement in treatment. With "*absolutely no way to read emotions or body language*" (22), these participants noted that the lack of face-to-face contact "*took away from the conversational ability of therapy, involving reactions and facial expression*" (110). One participant reflected how missing these essential interpersonal elements was "*almost more stressful than nothing at all. . .can't make eye contact over zoom, can't interpret silence*" (804). Authentic and face-to-face communication was clearly an integral component of therapy for many participants; and the lack of this, or a fragmented version thereof, proved consequential for their ability to contribute to the therapeutic alliance.

## Discussion

Amid the rapid and widespread integration of telemental health services in Australia during the unprecedented tenure of COVID-19, this study aimed to explore men's experiences with this emergent service modality. Relative to face-to-face care, younger, gay and university-educated men were all more likely to have engaged with any level of telemental health throughout the pandemic. On average, satisfaction with therapy was higher among those who had engaged with any telemental health than only face-to-face care. Specifically, among those who used telemental health, minimal differences were observed in our exploration of which men were most satisfied with this service; aside from expectedly lower satisfaction among those who experienced a lengthy wait to access therapy, and positive relationships between satisfaction with care and openness to receive telemental health in future. Qualitative responses regarding the benefits of telemental health for men revealed that this modality offered comfort, accessibility and convenience related to accessing therapy within a familiar environment (i.e., home), efficiently scheduling with other commitments (i.e., work) in order to accommodate face-to-face sessions. Notwithstanding this, several telemental health shortfalls were also noted, themed around therapy interrupted and disconnection with service providers, blurred boundaries between home and therapy and challenges with remote displays of emotional authenticity. In addition, many men felt telemental health was impersonal and lacked necessary elements of therapeutic connection provided by face-to-face attendance.

### Quantitative findings

Principally regarding use of telemental health services, our results derived from data collected almost two years into the pandemic suggest widespread uptake and norming of virtual therapy by men in the community. The majority of participants had some experience with telemental health; where among those, it was most common for participants to have all sessions via telemental health. Highlighted was a high proportion of the sample being open to receiving

telemental health in future; this finding differs slightly from results of Venville and colleagues [31] who observed a greater future preference for face-to-face services (55%) relative to telemental health only (15%). Given our data were collected nearly a year further into the pandemic, this difference could signal increasing normative framing and participants' openness to and high and rising need for mental healthcare. Nevertheless, many men's preference for face-to-face therapy highlights the need to improve distanced therapy experiences, particularly considering evidence that being male is associated with a poorer telemental health experience [6].

Our results demonstrated that relative to clients of only face-to-face therapy, those who had used telemental health were younger, more educated and more likely to identify as a gay. This aligns with findings of greater telemental health acceptability among younger people [32] and Isautier and colleagues' [6] assertions that university-educated people are more likely to use telemental health. Taken together this signals comfort with this technology with the potential to develop these resources with the ever-growing mental health needs of young men. Additionally, the use of telemental health was more common among gay men, and this might be explained as a by-product of sexual minority stresses that can restrain face-to-face mental health help-seeking [33]. The current study adds to gendered understandings of telemental health services in delineating sexual identity sub-groups who might draw additional benefits from accessing virtual help. This novel finding signals the resonance of telemental health among gay men, and points to further research that might usefully compare other treatment modalities to address the well-established barriers to help-seeking and mental illness risk in this sub-group [34].

By comparison, aside from lower satisfaction with telemental health among regional and rural participants relative to those living in urban areas, no sociodemographic variables emerged as significant factors differentiating levels of satisfaction with telemental health experiences. This is particularly encouraging given the widespread roll out of this service across Australia suggesting it might not need drastic tailoring for different populations. Conversely, an inverse relationship was observed between wait time to commence telemental health and subsequent satisfaction with care, reinforcing this delay as a significant barrier to engagement [35]. However, this finding is especially problematic given a common pattern among men to refrain from engaging mental healthcare until the point of crisis [17, 36]. Lower satisfaction with care among those who had to wait in excess of two months to commence therapy could therefore signal the influence of worsening, complex symptoms that may have been especially challenging for practitioners (and men) to respond to. These results highlight the importance of timely and targeted men's services and the potential use of adjunct online support during wait times to provide initial symptom management and/or coping skills training.

## Qualitative findings

Participants described how telemental health afforded them a greater sense of comfort and feeling of safety in allowing them to access care from a familiar environment, leading to more comfort disclosing distress. This is relative to the ever-present 'foreignness' of clinical mental health environments and practices which can invoke indebtedness and emasculating forces reinforcing the 'patient' narrative [37]. Accessing therapy remotely also carried the added bonus of mitigating risk of COVID-19 transmission inherent in face-to-face therapy; similar findings have been previously noted [38], and Isautier and colleagues [6] suggested face-to-face appointments were prone to cancellation due to fear of virus transmission. Improved and timely access to care in regional settings with greater choice for specialist care was also noted, reinforcing the value of telemental health in these settings where services are typically limited [39, 40].

By far the most commonly reported positive aspect of telemental health was the extent to which this modality allowed men to access therapy without compromising other commitments, primarily work. Importantly, structural and/or logistical barriers to care such as travel costs and the need to take time off work to access sessions have emerged as common factors impeding men's pathways to care [16, 21]. Current results suggest many of these barriers can be effectively ameliorated by telemental health, again allowing men opportunities to incorporate therapy into their lives with minimal scheduling impacts and reduced burdensomeness linked to needing to take time off work [41].

While for some, being able to engage in therapy from home was of benefit, for others this lack of separation between their everyday, private life and the therapeutic space was confusing and jarring. Ranging from difficulty finding a private area at home where they wouldn't be disturbed or overheard, to missing the utility of time and/or space for reflection therapy (e.g., in the commute to or from therapy) posed challenges for some men. As a result, the same technical limitations that plague all online activities were described here, a common finding in the recent telemental health literature [6, 31], with poor internet connection leading to frustration, and stymied emotional connections amid privacy concerns. This issue with connection extended beyond the internet to the space between therapist and client, and the most common concern raised was a feeling of impersonality and a difficulty building rapport in an online telemental health environment. As Venville and colleagues [31] suggest, face-to-face therapy affords a unique 'therapeutic holding' in a physical space that the current participants struggled to attain at a distance. Although in some cases this distance and sense of control can afford men the ability to disclose more freely, the majority spoke of missing the rich communicative and interpretive components of therapy, especially non-verbal cues including eye contact, visual cues and body language [6, 13]. Moreover, apparent in our results were nuances regarding the benefits and drawbacks of telemental healthcare when delivered via videoconference (e.g., Zoom) or via phone. We were unable to determine whether participants were referring to videoconference or phone-based appointments in qualitative results (which is discussed below as a limitation). Our results nevertheless suggest that perhaps there are unique considerations for therapists when engaging men via videoconference relative to phone-based sessions. Videoconference sessions may allow therapists greater depth of communication via non-verbal cues, however this modality presents more technological frustrations due to reliance on internet connectivity. Conversely, phone-based sessions may alleviate technological concerns, but participants lamented their impersonal nature due to the lack of visual feedback. When delivering therapy via telemental health, gauging client preferences and aiming to cater to these where options for modes of service provision are available, could boost likelihood of engagement.

## Future implications

Given the speed at which the mental health sector has been forced to integrate wide ranging telemental health services, it is possible that the examples of interpersonal and technical disconnection described above were more a reflection of clinician confidence or competence in remote rapport building, more than limitations of the telemental health modality itself. As has been noted previously, clinician doubt in the ethics and effectiveness of telemental health often outweighs that of clients [9, 32], and the broad acceptance of the need for telemental health in future among the current sample suggests that many men are open to this treatment style. It is clear therefore that best-practice guidelines and standards of care are required from peak bodies and government that can underpin telemental health-specific training programs and more rigorous research on when and how it is implemented to reliably improve treatment safety and

quality. It is imperative for practitioners to uphold their responsibility to appraise both the opportunities and drawbacks when delivering telemental health services among men, where often such idiosyncrasies as the blurring of boundaries between home and therapy can serve to distance clients [39]. When in therapy with male clients, evidence points to the importance of expectation setting to orient men to treatment [15]. This may be especially pressing when employing telemental health and thus purposefully clarifying its benefits and potential drawbacks is essential to create a shared understanding of the treatment journey.

## Limitations

The current study has several limitations of note. Firstly, the term telemental health may include phone or videoconference modalities, and this delineation was not explicitly made in the current survey, precluding us from understanding whether the present results pertain to telephone or videoconference services (or a combination of both). However, past evidence suggests satisfaction is comparable between the two [42]. Additionally, our primary comparison in this study was between participants who had *any* experience of telemental health, with those who *only* had face-to-face therapy in their most recent experience. Results should be considered in line with the fact that the telemental health group contains participants who received a mix of telemental health and face-to-face sessions in their most recent experience. We posit that this comparison nevertheless reflects the nature of the data which were collected across a period of time where many participants transitioned between telemental health and face-to-face therapy in tandem with ever-changing government social distancing mandates in Australia. Secondly, the design of this research was such that its quantitative findings are limited by its cross-sectional nature and the qualitative findings are limited by an inability to probe for elaboration. Moreover, while greater breadth is possible with the current methodology in gaining open responses from hundreds of participants, the structure of such fixed questioning can be presumptive and/or leading in nature. Ensuring future research expands on the male subgroup analyses described herein to explore why older, less educated or heterosexual men may comparatively be less likely to uptake telemental health is key. Further, the Australian-based context limits the generalisability of the current findings to other countries and contexts.

## Conclusion

The current findings provide important insights to men's telemental healthcare experiences within the COVID-19 context. While satisfaction was generally high among participants and most were willing to engage with it again in future, qualitative responses highlighted some elements around challenges building rapport that may guide adjustments both in terms of technological advances and telemental health specific training for mental health clinicians.

## Supporting information

**S1 Appendix. Survey items.**
(DOCX)

## Acknowledgments

The authors wish to thank all participants for their contribution to this study.

## Author Contributions

**Conceptualization:** Zac E. Seidler, John L. Oliffe, Simon M. Rice.

**Data curation:** Zac E. Seidler, Michael J. Wilson.

**Formal analysis:** Zac E. Seidler, Michael J. Wilson.

**Funding acquisition:** Zac E. Seidler.

**Methodology:** Michael J. Wilson, David Kealy.

**Supervision:** John L. Oliffe.

**Writing – original draft:** Zac E. Seidler, Michael J. Wilson.

**Writing – review & editing:** Zac E. Seidler, Michael J. Wilson, John L. Oliffe, David Kealy, John S. Ogrodniczuk, Andreas Walther, Simon M. Rice.

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
