## [Decision Letter · Decision Letter 0]

14 Jul 2022

PONE-D-22-08427“I could hang up if the practitioner was a prat”: Australian men’s feedback on telemental health care during COVID-19PLOS ONE

Dear Dr. Seidler,

Thank you for submitting your manuscript to PLOS ONE; I sincerely apologise for the unusually delayed review timeframe. Your manuscript has been assessed by one reviewer, whose comments are appended below. After careful consideration, we feel that it has merit but does not fully meet PLOS ONE’s publication criteria as it currently stands. Although the reviewer notes that the study "addresses an important public health issue" they raise a number of concerns that should be addressed, including around presentation of the data, and the discussion of the results. Therefore, we invite you to submit a revised version of the manuscript that addresses the points raised during the review process. Please note that we have only been able to secure a single reviewer to assess your manuscript. We are issuing a decision on your manuscript at this point to prevent further delays in the evaluation of your manuscript. Please be aware that the editor who handles your revised manuscript might find it necessary to invite additional reviewers to assess this work once the revised manuscript is submitted. However, we will aim to proceed on the basis of this single review if possible.

We look forward to receiving your revised manuscript.

Kind regards,

Emily Chenette

Editor in Chief

PLOS ONE

Journal Requirements:

Reviewers' comments:

Reviewer's Responses to Questions

**Comments to the Author**

1. Is the manuscript technically sound, and do the data support the conclusions?

Reviewer #1: Yes

2. Has the statistical analysis been performed appropriately and rigorously? 

Reviewer #1: I Don't Know

3. Have the authors made all data underlying the findings in their manuscript fully available?

Reviewer #1: No

4. Is the manuscript presented in an intelligible fashion and written in standard English?

Reviewer #1: Yes

5. Review Comments to the Author

Reviewer #1: This is a clear, well written paper that addresses an important public health issue: the acceptability to telemental health in men in Australia. As noted in the manuscript, this is important due to the high rates of suicides in Australian men and the exacerbation of factors associated with suicide due to COVID restrictions. The study is ‘mixed methods’ in that it employs a survey that has both quantitative and qualitative elements. I have a number of recommendations to make for the improvement of the paper.

1. While the authors reference Braun and Clarke for their qualitative analysis, qualitative findings were reported using both numbers and percentages. In general, Braun and Clarke advocate the use of terms such as ‘many, most, some’ rather than enumeration. Arguably when qualitative data is collected in a survey format it is appropriate to give a number to indicate the strength of a finding for comparative purposes, but I strongly recommend removing the percentages this suggest that respondents outside the percentage named did not support the particular response (whereas given the opened ended question, they may well support it but did not think of it at the time, or did not wish to fill out the qualitative fields).

2. The first qualitative theme emphasised comfort and safety, while the second emphasised convenience. I suggest that the finding regarding avoiding risk of COVID fits into the first theme, while, the convenience of being able to still do therapy if in isolation is appropriately in the second theme.

3. The fourth column on Table 1 is not clear. I suggest not using the three different indicators together in a single filed.

4. In the second and third columns in Table 1, I assume that the bracketed number is a percentage – this should be specified.

5. Table 2 is very confusing – is there a simple, clear, well labelled way to present this information?

6. PLOS authors have the option to publish the peer review history of their article (what does this mean?). If published, this will include your full peer review and any attached files.

Reviewer #1: **Yes: **Bridget Haire

---

## [Author Response · Author response to Decision Letter 0]

7 Aug 2022

1. While the authors reference Braun and Clarke for their qualitative analysis, qualitative findings were reported using both numbers and percentages. In general, Braun and Clarke advocate the use of terms such as ‘many, most, some’ rather than enumeration. Arguably when qualitative data is collected in a survey format it is appropriate to give a number to indicate the strength of a finding for comparative purposes, but I strongly recommend removing the percentages this suggest that respondents outside the percentage named did not support the particular response (whereas given the opened ended question, they may well support it but did not think of it at the time, or did not wish to fill out the qualitative fields). 

Thank you for raising this important point. We have followed your recommendation and removed all percentages from our reporting of the qualitative survey results. 

2. The first qualitative theme emphasised comfort and safety, while the second emphasised convenience. I suggest that the finding regarding avoiding risk of COVID fits into the first theme, while, the convenience of being able to still do therapy if in isolation is appropriately in the second theme.

 Thank you – we have moved the below sentence to the first theme as we agree it relates more to the first theme:

Eighteen participants described how this sense of comfort and control underpinned feelings of psychological and/or physical safety in the telemental health setting. The lower risk of contracting COVID-19 was relayed by some participants with “ensured protection” (369) and “health safety” (626), improving their comfort in accessing therapy in a pandemic context.

3. The fourth column on Table 1 is not clear. I suggest not using the three different indicators together in a single filed.

We have amended this column such that the significance values and effect size estimates are presented together in a separate column from the test statistics (i.e., chi square / t-values).

4. In the second and third columns in Table 1, I assume that the bracketed number is a percentage – this should be specified.

We have revised column 1 to make this clearer. The bracketed numbers in columns 2 and 3 are indicated by the terms in parentheses in the first column. That is, where Age: M (SD) appears in the first column, then the first value in column 2 is the mean, and the bracketed value is the standard deviation. 

5. Table 2 is very confusing – is there a simple, clear, well labelled way to present this information?

Table 2 contains the results from our multiple linear regression of satisfaction with therapy according to demographic and help-seeking predictors. We appreciate there are a range of statistical terms presented in this table, however we are limited in how to more concisely present this information. We have moved the confidence interval values for B to follow the B column as this makes more logical sense – hopefully this helps in our presentation of Table 2.

---

## [Decision Letter · Decision Letter 1]

31 Oct 2022

PONE-D-22-08427R1“I could hang up if the practitioner was a prat”: Australian men’s feedback on telemental health care during COVID-19PLOS ONE

Dear Dr. Seidler,

Thank you for submitting your manuscript to PLOS ONE. We apologise for the time this has taken in review.We feel that the paper has merit but requires some more minor revisions. Therefore, we invite you to submit a revised version of the manuscript that addresses the points raised during the review process.

We look forward to receiving your revised manuscript.

Kind regards,

Bridget Gabrielle Haire, PhD

Guest Editor

PLOS ONE

Journal Requirements:

Reviewers' comments:

Reviewer's Responses to Questions

**Comments to the Author**

1. If the authors have adequately addressed your comments raised in a previous round of review and you feel that this manuscript is now acceptable for publication, you may indicate that here to bypass the “Comments to the Author” section, enter your conflict of interest statement in the “Confidential to Editor” section, and submit your "Accept" recommendation.

Reviewer #2: (No Response)

2. Is the manuscript technically sound, and do the data support the conclusions?

Reviewer #2: Yes

3. Has the statistical analysis been performed appropriately and rigorously? 

Reviewer #2: I Don't Know

4. Have the authors made all data underlying the findings in their manuscript fully available?

Reviewer #2: No

5. Is the manuscript presented in an intelligible fashion and written in standard English?

Reviewer #2: Yes

6. Review Comments to the Author

Reviewer #2: Thank you for the opportunity to review this manuscript. Drawing on mixed methods research, this manuscript provides insights into the experiences of men accessing telemental health throughout the pandemic.

The authors have done well to address previous reviewer comments. I have a few additional comments.

Throughout:

health care and healthcare appear to be used interchangeably throughout

Numbers below 10 should be written out (e.g., two, not 2)

Abstract:

How were these men recruited?

Introduction:

Page 3, lines 80-81: Feels like a qualifier is needed here. One in five Australian adults?

Page 5, line 107: should this be “accompanying the COVID-19 pandemic”?

Method:

Page 6, line 129: Why 16+, rather than say, 18+?

Page 7, lines 160-161: was there much turnover of practitioner (and was this information collected)?

Results:

page 12, line 240: insert full stop

page 15, line 261 “on their own terms”: not clear what is meant by this - was it that they had the choice of attending in person or via telehealth? Or something else?

Page 15, quote in lines 264-265: Is there more to this quote? It would seem that a therapist's office would likely have fewer external distractions than someone's home.

Page 16, lines 267-270 (quote): the quote seems more suited to the next sub-section – “Convenience and accessibility”

Pages 16-17, Lines 285 (“physical distance”) – 293: the next two quotes suggest it isn't the physical distance, but the anonymity of the phone (vs video telehealth sessions, such as over Zoom or FaceTime) that they appreciate.

Page 18, line 327: should “regards” be “regarding” or “with regards to”?

Page 18, line 330: change "the frustrating" to "some frustrations with the". The way it is currently written suggests that the technical limitations are inherently frustrating, rather than only being frustrating for some people in some instances.

Page 19, line 343 “some”: Why is "some" used here when other responses are quantified with exact number of participants who relayed similar responses?

Page 19, lines 345-356: This paragraph / section might be better placed after the one describing the pros of telehealth (can stay in pjs, don't need to travel, etc).

page 22, line 402: “challenges being emotionally authentic remotely.” – is there a word missing?

page 25, line 469 “utility of reflection”: is something missing here? e.g., utility of time/space for reflection, such as car drive home?

Limitations

Page 27, lines 500-504: I think greater mention and exploration of this is needed in the discussion, particularly as the nuanced experiences of both are apparent in the data and results (e.g., page 19, line 348 – participant 179 quote and page 20, line 378, participant 110).

7. PLOS authors have the option to publish the peer review history of their article (what does this mean?). If published, this will include your full peer review and any attached files.

Reviewer #2: No

---

## [Author Response · Author response to Decision Letter 1]

7 Nov 2022

Thank you to our reviewer for taking the time to review our manuscript. We have addressed all comments in the response table attached with this submission, and hope the manuscript is now suitable for publication. Please let us know if there is anything else you require. 

Kind regards,

Dr Zac Seidler

---

## [Decision Letter · Decision Letter 2]

21 Nov 2022

PONE-D-22-08427R2“I could hang up if the practitioner was a prat”: Australian men’s feedback on telemental health care during COVID-19PLOS ONE

Dear Dr. Seidler,

Thank you for submitting your manuscript to PLOS ONE. After careful consideration, we feel that it has merit but does not fully meet PLOS ONE’s publication criteria as it currently stands. Therefore, we invite you to submit a revised version of the manuscript that addresses the points raised during the review process.

We look forward to receiving your revised manuscript.

Kind regards,

Nabeel Al-Yateem, PhD

Academic Editor

PLOS ONE

Journal Requirements:

Reviewers' comments:

Reviewer's Responses to Questions

**Comments to the Author**

1. If the authors have adequately addressed your comments raised in a previous round of review and you feel that this manuscript is now acceptable for publication, you may indicate that here to bypass the “Comments to the Author” section, enter your conflict of interest statement in the “Confidential to Editor” section, and submit your "Accept" recommendation.

Reviewer #2: All comments have been addressed

2. Is the manuscript technically sound, and do the data support the conclusions?

Reviewer #2: Yes

3. Has the statistical analysis been performed appropriately and rigorously? 

Reviewer #2: I Don't Know

4. Have the authors made all data underlying the findings in their manuscript fully available?

Reviewer #2: No

5. Is the manuscript presented in an intelligible fashion and written in standard English?

Reviewer #2: Yes

6. Review Comments to the Author

Reviewer #2: The authors have fully addressed all comments. I appreciate the explanations provided in instances where reviewer comments/suggestions were not implemented.

My last remaining suggestion would be a sentence in the Methods "Experiences with telemental health" subsection, to explicitly state that in the survey, the term "telemental health" referred to both virtual (video) and phone (audio only). As this is addressed in the limitations, I think it is important for the reader to have this information up front prior to reading and interpreting the results.

7. PLOS authors have the option to publish the peer review history of their article (what does this mean?). If published, this will include your full peer review and any attached files.

Reviewer #2: No

---

## [Author Response · Author response to Decision Letter 2]

27 Nov 2022

Thank you for the comments on our revision. We have addressed the final concern in the attached response letter.

---

## [Editor Report · Decision Letter 3]

1 Dec 2022

“I could hang up if the practitioner was a prat”: Australian men’s feedback on telemental health care during COVID-19

PONE-D-22-08427R3

Dear Dr. Seidler,

We’re pleased to inform you that your manuscript has been judged scientifically suitable for publication and will be formally accepted for publication once it meets all outstanding technical requirements.

Kind regards,

Nabeel Al-Yateem, PhD

Academic Editor

PLOS ONE
---

## [Editor Report · Acceptance letter]

5 Dec 2022

PONE-D-22-08427R3 

*“I could hang up if the practitioner was a prat”*: Australian men’s feedback on telemental healthcare during COVID-19   

Dear Dr. Seidler:

I'm pleased to inform you that your manuscript has been deemed suitable for publication in PLOS ONE. Congratulations! Your manuscript is now with our production department. 

Kind regards, 

on behalf of

Dr. Nabeel Al-Yateem 

Academic Editor

PLOS ONE